# Detection of Meat Adulteration Using Spectroscopy-Based Sensors

**DOI:** 10.3390/foods10040861

**Published:** 2021-04-15

**Authors:** Lemonia-Christina Fengou, Alexandra Lianou, Panagiοtis Tsakanikas, Fady Mohareb, George-John E. Nychas

**Affiliations:** 1Laboratory of Microbiology and Biotechnology of Foods, Department of Food Science and Human Nutrition, School of Food and Nutritional Sciences, Agricultural University of Athens, 11855 Athens, Greece; p.tsakanikas@aua.gr (P.T.); gjn@aua.gr (G.-J.E.N.); 2Division of Genetics, Cell Biology and Development, Department of Biology, University of Patras, 26504 Patras, Greece; 3Bioinformatics Group, Department of Agrifood, School of Water, Energy and Environment Cranfield University, College Road, Cranfield, Bedfordshire MK43 0AL, UK; f.mohareb@cranfield.ac.uk

**Keywords:** adulteration, minced meat, multispectral imaging, sensors, spectroscopy

## Abstract

Minced meat is a vulnerable to adulteration food commodity because species- and/or tissue-specific morphological characteristics cannot be easily identified. Hence, the economically motivated adulteration of minced meat is rather likely to be practiced. The objective of this work was to assess the potential of spectroscopy-based sensors in detecting fraudulent minced meat substitution, specifically of (i) beef with bovine offal and (ii) pork with chicken (and vice versa) both in fresh and frozen-thawed samples. For each case, meat pieces were minced and mixed so that different levels of adulteration with a 25% increment were achieved while two categories of pure meat also were considered. From each level of adulteration, six different samples were prepared. In total, 120 samples were subjected to visible (Vis) and fluorescence (Fluo) spectra and multispectral image (MSI) acquisition. Support Vector Machine classification models were developed and evaluated. The MSI-based models outperformed the ones based on the other sensors with accuracy scores varying from 87% to 100%. The Vis-based models followed in terms of accuracy with attained scores varying from 57% to 97% while the lowest performance was demonstrated by the Fluo-based models. Overall, spectroscopic data hold a considerable potential for the detection and quantification of minced meat adulteration, which, however, appears to be sensor-specific.

## 1. Introduction

In the last decade food fraud has become a major issue, which, among others, is reflected on the exponentially increasing number of pertinent studies published in the scientific literature (Figure 1). Food fraud scandals such as milk adulterated with melamine in China (2008), horsemeat in beef products (2013) and fipronil in eggs (2017) have had a negative credibility and financial impact affecting authorities, industries and consumers worldwide [1]. Although the motivation of most fraudulent practices is economic gain, the consequences are not only restricted to the deception of consumers but may also affect public health (e.g., allergies or food poisoning) or lead to the consumption of undesirable food commodities for religious or cultural reasons [2,3]. Food fraud is a very challenging issue to deal with because of the complexity of the global food supply chain as well as the unknown time, practice and place of the conducted fraudulent practice. Hence, food fraud mitigation, especially through the prevention of such practices by the implementation of frequent inspections, is of major importance.

The world meat output in carcass weight was 338.8 million tons in 2019 and the highest produced meat categories were bovine, poultry and pig [4]. Various minced meat products are available in the market including meatballs, hamburgers, sausages and frozen or ready-to-eat meals. The high demand of meat products in conjunction with unfair trade render minced meat susceptible to adulteration, which is also alleviated by the fact that it can be easily masked. Certainly, the morphological properties of intact meat are not present in minced meat and thus the partial substitution of one meat type with a cheaper alternative species or tissues can be readily concealed. For example, offal may be regarded as both unpalatable and a delicacy depending on social, cultural and lifestyle aspects. Actually, offal consumption is not popular in most developed countries [5]. Nonetheless, offal could be used as an adulterant of costly meat products with the ultimate goal of economic gain. It is worth mentioning that the meat and poultry food categories were among the top five product categories with the most requests concerning the suspicion of fraud within the EU Administrative Assistance and Cooperation (AAC) System in the annual report of 2019 [6]. In this framework, the detection of non-compliance via regular inspections followed by their further characterization as accidental or fraudulent are of vital importance for food fraud control and mitigation.

The rapid and reliable detection of minced meat adulteration is of apparent value in terms of food protection throughout the food supply chain. Various methods have been investigated for meat authentication and also for the detection of adulteration such as electrophoretic, enzymatic, chromatographic and spectroscopic methods [7,8,9]. Analytical methods for the detection of meat adulteration are often based on protein and DNA analysis [10,11]. According to the European Parliament Resolution of 14 January 2014, DNA testing is suggested as a standard procedure for determining animal species for the purpose of fraud detection and control [12]. However, this method is time-consuming, expensive, invasive and requires highly trained personnel. On the other hand, spectroscopy-based methods have gained the interest of researchers because of their rapid, non-invasive character and the potential to be applied on, in and at line [13,14]. Spectroscopic techniques such as near-infrared (NIR), mid-infrared (MIR), ultraviolet-visible (UV-Vis), Raman and laser-induced breakdown spectroscopy (LIBS) [15,16] as well as multispectral or hyperspectral imaging (MSI, HSI) methods [17,18], in tandem with machine learning approaches, have been investigated for: (i) meat species identification; (ii) the detection of animal or plant origin adulterants in meat and meat products; (iii) the detection of frozen-thawed meat and the detection of other food fraud cases [19,20,21,22,23,24,25,26].

The adulteration of beef with bovine offal could be practiced due to the relatively low price of the latter stemming from its low popularity among consumers. The adulteration of pork with chicken could also be practiced because pork is usually more expensive than chicken. Nonetheless, the adulteration of chicken with pork is also likely because, under certain conditions, this may also constitute an economically motivated activity. Indeed, organic or free-range chicken products, which are increasingly gaining popularity among consumers, commonly have an equal/higher price to/than pork, while specific chicken cuts (e.g., chicken breasts) are also more expensive than specific pork meat cuts (e.g., pork shoulder). Moreover, the possible economically motivated adulteration scenarios are limitless because they can be driven by parameters other than the meat type/cut; for example, pork meat close to an expiration date could be used as an adulterant in a chicken product. Finally, the detection of pork in products commercialized as chicken products is also important for the mitigation of potential health risks (e.g., allergies) as well as for the protection of consumer rights with regard to religious or cultural aspects.

In this context, the aim of the present study was the investigation of the efficacy of spectroscopy-based sensors in the detection of meat adulteration practiced through the partial substitution of a claimed meat with meat from another animal species (i.e., pork with chicken and vice versa) or with another tissue of the same species (i.e., beef with bovine offal). An additional objective of this work was to assess whether the adulteration detection efficacy of spectroscopy technologies was similar between fresh and frozen-thawed meat samples. In this context, two scenarios of minced meat adulteration were studied; namely, (i) the adulteration of pork with chicken (and vice versa) and (ii) the adulteration of beef with bovine offal.

## 2. Materials and Methods

### 2.1. Sample Preparation

The experimental setup was applied for each case of meat adulteration, i.e., pork with chicken (and vice versa) and beef with bovine offal (bovine hearts). The meat/offal pieces were purchased from four different butcher shops in Athens, Greece (b1, b2, b3, b4) and were ground separately (one type of meat at a time) using a domestic meat-mincing machine. The parts of the machine were cleaned and the procedure was repeated for the other types of meat. To achieve different levels of adulteration with a 25% step, appropriate portions of meat were mixed. For every level (0%, 25%, 50%, 75%, 100%), six Petri dishes (i.e., sample replicates of ca. 70–80 g each) were prepared. The experiment was repeated four times. In total, 120 samples (5 adulteration levels × 6 samples × 4 batches) were prepared for each meat adulteration scenario. The meat samples were frozen (−20 °C) and after ca. 14 weeks were thawed (6–8 h incubation at 4 °C).

### 2.2. Measurements Using Spectroscopy-Based Sensors and Data Acquisition

The data acquired for fresh samples were: (i) visible (Vis) spectra, (ii) fluorescence (Fluo) signals and (iii) multispectral imaging (MSI) data while MSI was also used for the frozen-thawed samples (due to its outperformance compared with the other sensors as will be discussed in Section 3 and Section 4).

#### 2.2.1. Vis and Fluo Data

The UV-Vis spectrometer Hamamatsu C12880MA (Hamamatsu Photonics K.K., Shizuoka, Japan) has a spectral range from 340 to 850 nm and a resolution of 15 nm. The sensor was employed for both Vis spectroscopy and the detection of Fluo signals. For each meat sample, 10 spectral measurements (different spots on the sample) were acquired using the Vis and Fluo modes of the sensor and the average value was used for the data analysis [27].The visible regions of 430–710 and 400–800 nm were used for the beef-offal and pork-chicken adulteration scenarios, respectively, while the corresponding regions for the analysis of the Fluo signals were 384–496 and 384–564 nm. The regions that were excluded were due to high noise levels at the specific wavelengths. The differences between the two use cases (beef-offal and pork-chicken) can be attributed to the different physiochemical properties of the raw materials in relation to the incident radiation.

#### 2.2.2. MSI Data

Data were acquired using the VideometerLab system originally developed by the Technical University of Denmark [28] and commercialized by “Videometer A/S” [29]. This instrument acquires multispectral images in 18 different, non-uniformly distributed wavelengths ranging from Vis (405 nm) to short wave NIR (970 nm). The acquired images were subjected to segmentation so that only the informative region (muscle tissue) was included for the calculation of the mean reflectance spectrum (i.e., mean intensity of pixels within the informative area) along with the corresponding standard deviation values. The instrument and the data extraction procedures have been described in detail previously [30]. Beyond the spectral information also provided by the UV-Vis spectrometer, MSI data combined both spectral and spatial information.

### 2.3. Data Analysis

The models were developed using three independent batches (b1, b2, b3) and validated with an external batch (b4) so that a 75–25% proportion of the whole dataset was used for training-testing (external validation). The developed models were based on the algorithm of the support vector machines (SVMs), which is referred to as one of the most popular algorithms for meat species identification by Kumar and Karne [15]. The X-variables were transformed using the partial least-squares (PLS) algorithm [31,32] prior to the development of the classification models. PLS is a method for relating two data matrices (X, Y) and can be employed for dimensionality reduction and extraction. Downstream in the analysis pipeline, the number of PLS components explaining more than 96% of the input data variance and less than three components were used. In the case that more than three components were needed to explain 96% of the variance, only three components were used. Subsequently, the algorithm of the SVMs was applied on the transformed and redundant variables [33,34]. Intuitively, the algorithm of the SVMs maps the original data to a high-dimensional feature space using a kernel function so that a maximal separating hyperplane can be constructed. SVMs with a radial basis function (RBF) kernel were employed in the present study. A grid search in tandem with a 10-fold cross validation was performed to the training set for tuning the parameters of cost and gamma. The experimental setup and applied data analysis are shown in Figure 2.

The SVMs and PLS were implemented in R (R Foundation for Statistical Computing, version 4.0.3, Vienna, Austria) [35] and Rstudio (RStudio, version 1.3.1093, Boston, Massachusetts, USA) [36] using the package ‘e1071′ [37] and ‘pls’ [38], respectively. Pork-chicken and beef-offal adulteration scenarios were investigated for the classification potential among different adulteration levels (5-class model) and for the discrimination of pure types from adulterated meat samples (3-class model). The models’ performance was evaluated by calculating accuracy, kappa (a measure that compares an observed accuracy with an expected accuracy (random chance)), specificity, recall, precision and F1-score, which is the harmonic mean of the precision and recall of a specific class. The confusion matrices of each examined case and sensor are available in the Appendix A. In the present study, a multi-class (3-class and 5-class) problem was faced and the performance metrics were calculated using the samples of one class (e.g., 1, 2, 3… *i*) against the samples of the other classes. The metrics and confusion matrices were calculated using the package ‘caret’ in R [39].
(1)$$\mathrm{Accuracy}\;=\frac{\mathrm{Samples}\;\mathrm{correctly}\;\mathrm{predicted}}{\mathrm{All}\;\mathrm{samples}}$$
(2)$$\mathrm{Kappa}\;=\frac{\mathrm{Observed}\;\mathrm{accuracy}-\mathrm{Expected}\;\mathrm{accuracy}}{1-\mathrm{Expected}\;\mathrm{accuracy}}$$
(3)$$\mathrm{Recall}\;=\frac{\mathrm{Samples}\;\mathrm{correctly}\;\mathrm{predicted}\;\mathrm{in}\;i\;\mathrm{class}}{\mathrm{All}\;\mathrm{samples}\;\mathrm{of}\;\mathrm{the}\;i\;\mathrm{class}}$$
(4)$$\mathrm{Precision}\;=\frac{\mathrm{Samples}\;\mathrm{correctly}\;\mathrm{predicted}\;\mathrm{in}\;i\;\mathrm{class}}{\mathrm{All}\;\mathrm{samples}\;\mathrm{predicted}\;\mathrm{as}\;i\;\mathrm{class}\;(\mathrm{correct}\;\mathrm{or}\;\mathrm{not})}$$
(5)$$\mathrm{Specificity}\;=\frac{\mathrm{Samples}\;\mathrm{correctly}\;\mathrm{predicted}\;\notin \;\mathrm{in}\;i\;\mathrm{class}}{\left(\mathrm{Samples}\;\mathrm{correctly}\;\mathrm{predicted}\;\notin \;\mathrm{in}\;i\;\mathrm{class})+(\mathrm{Samples}\;\mathrm{incorrectly}\;\mathrm{predicted}\;\in \;\mathrm{in}\;i\;\mathrm{class}\right)}$$
(6)$$F1-\mathrm{score}=\frac{2\;*\;(\mathrm{Recall}\;*\;\mathrm{Precision})}{\left(\mathrm{Recall}\;+\;\mathrm{Precision}\right)}$$

## 3. Results

### 3.1. Pork-Chicken Adulteration Scenario

Table 1 presents the performance metrics for the external validation and the classification in five classes for the different spectral data (Vis, MSI) and the different conditions of the samples (fresh, frozen-thawed). The MSI data in both cases of fresh and frozen-thawed samples yielded higher accuracies; namely, 90.00% and 86.67%, respectively, compared with the Vis data (fresh samples, accuracy = 56.67%). In the case of the classification models applied on Vis data on the SVMs, neither pure categories nor adulterated samples were distinguished easily with the attained F1-scores varying from 0.50 (class: 0% (pure chicken), 25%) to 0.67 (class: 75%). Contrary to the Vis data, the models developed using MSI data either in the case of fresh or in the case of frozen-thawed samples yielded higher performances especially the classes containing higher proportions of chicken (classes 0 and 25%), which yielded F1-scores = 1.00. Specificity scores for MSI data (fresh and frozen-thawed samples) were also over 87.50% but for Vis data were over 58.33%. In all cases (Vis, MSI) for classes 0% and 25% specificity scores were 100.00%. The misclassifications for the external validation set occurred in adjacent classes (± 1 classes) for the models trained using MSI data but for models trained using Vis data misclassifications occurred within ± 2 classes, as shown in the confusion matrices in the Appendix A.

Table 2 presents the respective performance metrics and cases for the classification in three classes. The classification of samples in three classes for the discrimination of pure from adulterated samples showed a higher potential with accuracy scores over 93.33% compared with the examined 5-class classification. In the case of the Vis spectral data, the accuracy of the classification in three classes was improved (i.e., 93.33%) compared with the attained accuracy in the 5-class classification (i.e., 56.67%). In the latter case, only two pure samples from the 100% class (Appendix A) were misclassified as adulterated (class: A). The observed misclassifications were a failsafe with the precision in both pure types (0%, 100%) being 100%. The same accuracy (i.e., 93.33%) was also attained when SVMs were applied on MSI data in the case of frozen-thawed samples but in this case, two adulterated samples were misclassified as pork (class: 100%) (precision = 75.00%; Appendix A). The classification models of SVMs applied on the MSI data for fresh samples were accurately distinguished with the attained accuracy being 100.00% (Appendix A). The Fluo data had poor performance. The confusion matrices are available in Appendix A. The accuracy scores were 60.00% (5-class) and 80.00% (3-class) and none of the pork samples (class: 100%) were classified correctly with a recall of 0.00% in both classification models.

### 3.2. Beef-Offal Adulteration Scenario

The classification models of SVMs for the detection of the adulteration of beef with bovine offal (bovine hearts) showed higher or equal performance in terms of accuracy scores for the respective cases compared with the pork-chicken adulteration scenario. Table 3 presents the performance metrics for the external validation and the classification in the five classes. MSI data in both cases of fresh and frozen-thawed samples were totally discriminated (accuracy = 100.00%). In the case of the classification models of SVMs applied on the Vis data, pure categories yielded F1-scores = 1.00 but all adulterated levels (i.e., classes: 25%, 50%, 75%) yielded F1-scores lower than 0.91. In the case of the middle class (50%), none of the samples were correctly classified (recall = 0.00%) and all of the misclassifications were noted in adjacent classes (±2) (Appendix A).

Table 4 shows the performance metrics for the classification in three classes. In all of the examined cases (Vis, MSI), the misclassifications were a failsafe where samples from pure types were misclassified as adulterated (Appendix A) (specificity = 100%). It is worth pointing out that the samples from the pure beef class were totally discriminated with F1-score = 1.00 for both sensors, which equaled total discrimination of the pure samples as long as the bovine offal was considered to be the adulterant. Models developed and evaluated using MSI data from frozen-thawed samples were totally discriminated with an accuracy of 100.00% (Appendix A). Poor performance was attained when Fluo data were used for the detection of adulteration (as shown in Appendix A) for classification in five and three classes, respectively. Accuracy scores were 50.00% (5-class) and 63.33% (3-class) with the latter accuracy indicating that there was no information rate and the samples were classified by chance. The misclassifications occurred within ± 5 classes, which indicated the poor fitting of the model (Appendix A).

Overall, when comparing Vis and MSI data for the detection of fresh samples, the lowest performance was attained when the algorithm of the SVMs was applied on Vis data for the development of 5-class classification models with kappa of 0.46 (chicken-pork) and 0.62 (beef-offal). In all other cases (3-class, 5-class), the kappa scores varied from 0.81 to 1.00 and could be characterized as a perfect agreement according to the arbitrary groups defined by Landis and Koch [40]; the proposed groups are the most common groups in literature for the interpretation of kappa [41]. Classification in three classes of the Vis data improved the kappa scores from 0.46 (5-class) to 0.87 (3-class) and from 0.62 (5-class) to 0.94 (3-class) for the pork-chicken and beef-offal adulteration scenarios, respectively. The models’ performance using MSI data for fresh and frozen-thawed samples was higher compared with the other two types of spectral data with kappa varying from 0.83 to 1.00. In the pork-chicken adulteration scenario, the performance (kappa) of frozen samples was better compared with the fresh samples as opposed to the case of the beef-offal adulteration scenario where the developed models based on the fresh samples were better performing than the ones based on the frozen-thawed samples (5-class) or equally performing in the case of the 3-class classification (100.00%). Although the Vis and MSI data covered similar spectral regions (visual spectrum), imaging methods (e.g., MSI) were applied on the whole region of interest (whole sample) compared with the specific point measurements provided by the UV/Vis spectrometer. Thus, apart from the spectral information, MSI also provided spatial information and, more importantly, it allowed for the representation of the whole sample instead of an arbitrary number of sites/points and, hence, a better assessment of the adulteration events. It was apparent that the region/surface of the samples used for further analysis affected the performance of the models particularly in terms of quantifying the proportion of the adulterants in the sample (5-class). The models based on Fluo data exhibited poorer performance compared with the other two spectral data types.

## 4. Discussion

There are many studies illustrating the potential of spectroscopic methods coupled with machine learning approaches in detecting meat adulteration. Nevertheless, the results obtained in the present study could not be easily compared with those of other studies because such results are sensor, algorithm and meat-specific. Promising results have been attained for the discrimination of pork from chicken using HSI (NIR) and MSI (Vis-NIR) [42,43]. Furthermore, several studies have been conducted for the discrimination of beef from bovine offal using spectroscopic methods. Black et al. [19] investigated the rapid detection of offal within minced beef samples utilizing ambient mass spectrometry with an 1–10% adulteration level being detectable. Laser-induced breakdown spectroscopy (LIBS) has been investigated as a potential indicator of offal adulteration in minced beef samples with high (0.85–0.95) values of coefficients of determination being reported for validation datasets [44,45]. Furthermore, vibrational spectroscopy has been investigated for offal detection in beef burgers using FTIR, Raman and NIR spectroscopies [46,47,48]. Morsy and Sun [49] also used NIR spectroscopy for the detection of adulterants in fresh and frozen-thawed beef samples; in the latter study, bovine offal was included as an adulterant but, in contrast to the findings of the present study, the detection of the frozen-thawed adulteration was slightly inferior compared with the detection of the adulteration in the fresh samples most likely due to the different sensors used. The detection of adulteration in frozen-thawed meat samples has been investigated for various scenarios such as turkey in beef and offal in beef burgers as well as for the discrimination of red meat (lamb, pork, beef) and promising results were attained [25,47,50]. As mentioned above, considerable potential has been exhibited for the detection of the adulteration of beef with bovine offal but, to the best of our knowledge, there are no studies utilizing MSI or HSI data for the detection of such adulteration in both fresh and frozen-thawed samples. It is worth mentioning that MSI and HSI methods have been applied in many cases for the purpose of meat fraud detection. Certainly, MSI or HSI data have been used for the detection of adulteration of beef with pork, beef with horsemeat, chicken with beef and duck with lamb meat [18,51,52,53]. The investigated cases of meat adulteration and the performance accomplished demonstrate that there is a great potential for the detection of meat adulteration using similar approaches.

The poor performance of Fluo data compared with the other spectral data (MSI, Vis) may be attributed to the food matrix, which leads to distortion (absorption and scattering) of the exciting light as well as to the reabsorption of the fluorescent light. It is also possible that the acquired Fluo data did not have enough informative power for the discrimination of the examined adulteration cases. It is worth mentioning that the limited studies on fluorescence spectroscopy for the purpose of authentication [54] and more specifically for meat adulteration issues makes it difficult to draw firm conclusions [55]. Further investigation using different modes of fluorescence spectroscopy and different adulteration cases is expected to provide more insight regarding the potential of this spectroscopy in contributing to the detection of meat adulteration.

The obtained results are important but further investigation is needed for implementing the proposed methods in real life. For example, model robustness is anticipated to be enhanced through the investigation of an increased number of samples and thus the acquisition of larger datasets, allowing for variations associated with both the food commodity and the adulterants to be taken into account. In this context, the inclusion of samples from different seasons, storage conditions and locations should be ensured [56]. Moreover, the development of general methods, the integration of different meat types and alternative machine learning approaches would allow for a holistic and realistic approach with regard to adulteration detection even in cases where no information regarding the encountered adulteration is available [57].

## 5. Conclusions

Spectroscopic data in tandem with machine learning approaches hold considerable potential for the detection and quantification of minced meat adulteration, which, however, appears to be sensor-specific. The highest performance was exhibited by the MSI-based models followed by the Vis-based models with the models based on Fluo spectral data demonstrating the lowest potential for the detection of meat adulteration.

## Figures and Tables

**Figure 1 foods-10-00861-f001:**
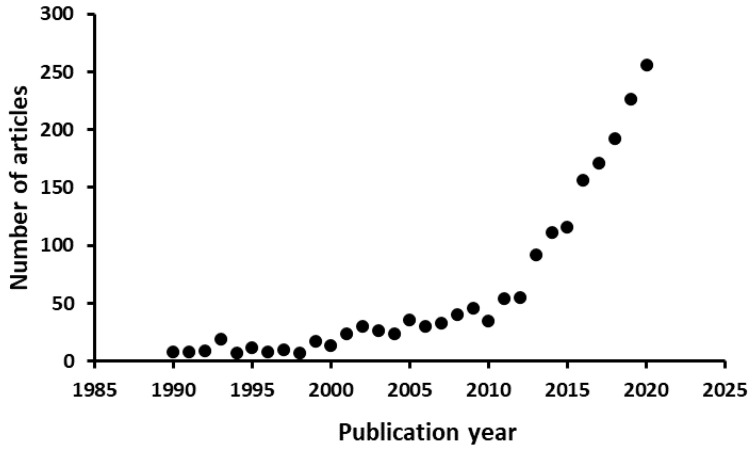
Number of published documents in Scopus from 1990 to 2020 with the term ‘food fraud’ within the article title, abstract and/or keywords.

**Figure 2 foods-10-00861-f002:**
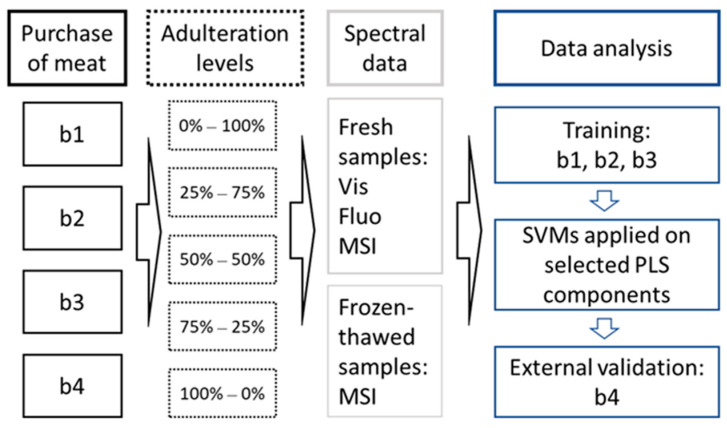
Experimental set up and data analysis for the two adulteration scenarios, i.e., pork with chicken (and vice versa) and beef with bovine offal. b: batch; Vis: visible spectra; Fluo: fluorescence signals; MSI: multispectral imaging; SVMs: support vector machines; PLS: partial least-squares.

**Table 1 foods-10-00861-t001:** Specificity, recall, precision, F1-score, accuracy and kappa for the classification of SVMs for the external validation (*n* = 30) of fresh samples using Vis and MSI data and of frozen-thawed samples using MSI data considering five classes from 0% pork-100% chicken (0%) to 100% pork-0% chicken (100%).

		True Class				
Sensors		0%	25%	50%	75%	100%
Vis fresh samples	Specificity (%)	100.00	100.00	58.33	91.67	95.83
	Recall (%)	33.33	33.33	100.00	66.67	50.00
	Precision (%)	100.00	100.00	37.50	66.67	75.00
	F1-score	0.50	0.50	0.54	0.67	0.60
	Accuracy (%)	56.67				
	Kappa	0.46				
MSI fresh samples	Specificity (%)	100.00	100.00	100.00	87.50	100.00
	Recall (%)	100.00	100.00	100.00	100.00	50.00
	Precision (%)	100.00	100.00	100.00	66.67	100.00
	F1-score	1.00	1.00	1.00	0.80	0.67
	Accuracy (%)	90.00				
	Kappa	0.87				
MSI frozen-thawed samples	Specificity (%)	100.00	100.00	100.00	95.83	87.50
	Recall (%)	100.00	100.00	83.33	50.00	100.00
	Precision (%)	100.00	100.00	100.00	75.00	66.67
	F1-score	1.00	1.00	0.91	0.60	0.80
	Accuracy (%)	86.67				
	Kappa	0.83				

**Table 2 foods-10-00861-t002:** Specificity, recall, precision, F1-score, accuracy and kappa for the classification of SVMs for the external validation (*n* = 30) of fresh samples using Vis and MSI data and of frozen-thawed samples using MSI data considering three classes: 0% pork-100% chicken (0%); adulterated (A); 100% pork-0% chicken (100%).

		True Class		
Sensors		0%	A	100%
Vis fresh samples	Specificity (%)	100.00	83.33	100.00
	Recall (%)	100.00	100.00	66.67
	Precision (%)	100.00	90.00	100.00
	F1-score	1.00	0.95	0.80
	Accuracy (%)	93.33		
	Kappa	0.87		
MSI fresh samples	Specificity (%)	100.00	100.00	100.00
	Recall (%)	100.00	100.00	100.00
	Precision (%)	100.00	100.00	100.00
	F1-score	100.00	100.00	100.00
	Accuracy (%)	100.00		
	Kappa	1.00		
MSI frozen-thawed samples	Specificity (%)	100.00	100.00	91.67
	Recall (%)	100.00	88.89	100.00
	Precision (%)	100.00	100.00	75.00
	F1-score	1.00	0.94	0.86
	Accuracy (%)	93.33		
	Kappa	0.89		

**Table 3 foods-10-00861-t003:** Specificity, recall, Precision, F1-score, accuracy and kappa for the classification of SVMs for the external validation (*n* = 30) of fresh samples using Vis and MSI data and of frozen-thawed samples using MSI data considering five classes from 0% beef-100% offal (0%) to 100% beef-0% offal (100%).

		True Class				
Sensors		0%	25%	50%	75%	100%
Vis fresh samples	Specificity (%)	100.00	70.83	100.00	100.00	100.00
	Recall (%)	100.00	100.00	0.00	83.33	100.00
	Precision (%)	100.00	46.15	^1^ NaN	100.00	100.00
	F1-score	1.00	0.63	^1^ NaN	0.91	1.00
	Accuracy (%)	76.67				
	Kappa	0.62				
MSI fresh samples	Specificity (%)	100.00	100.00	100.00	100.00	100.00
	Recall (%)	100.00	100.00	100.00	100.00	100.00
	Precision (%)	100.00	100.00	100.00	100.00	100.00
	F1-score	1.00	1.00	1.00	1.00	1.00
	Accuracy (%)	100.00				
	Kappa	1.00				
MSI frozen-thawed samples	Specificity (%)	100.00	100.00	100.00	100.00	100.00
	Recall (%)	100.00	100.00	100.00	100.00	100.00
	Precision (%)	100.00	100.00	100.00	100.00	100.00
	F1-score	1.00	1.00	1.00	1.00	1.00
	Accuracy (%)	100.00				
	Kappa	1.00				

^1^ NaN: Not a Number (i.e., the outcome of division by 0).

**Table 4 foods-10-00861-t004:** Specificity, recall, precision, F1-score, accuracy and kappa for the classification of SVMs for the external validation (*n* = 30) of fresh samples using Vis and MSI data and of frozen-thawed samples using MSI data considering three classes: 0% beef-100% offal (0%); adulterated (A); 100% beef-0% offal (100%).

		True Class		
Sensors		0%	A	100%
Vis fresh samples	Specificity (%)	100.00	91.67	100.00
	Recall (%)	83.33	100.00	100.00
	Precision (%)	100.00	94.74	100.00
	F1-score	0.91	0.97	1.00
	Accuracy (%)	96.67		
	Kappa	0.94		
MSI fresh samples	Specificity (%)	100.00	83.33	100.00
	Recall (%)	100.00	100.00	66.67
	Precision (%)	100.00	90.00	100.00
	F1-score	1.00	0.95	0.80
	Accuracy (%)	93.33		
	Kappa	0.87		
MSI frozen-thawed samples	Specificity (%)	100.00	100.00	100.00
	Recall (%)	100.00	100.00	100.00
	Precision (%)	100.00	100.00	100.00
	F1-score	1.00	1.00	1.00
	Accuracy (%)	100.00		
	Kappa	1.00		

## Data Availability

The data presented in this study are available on request from the corresponding authors. The data are not publicly available due to privacy restrictions.

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
