# Peer review of "Detection of Meat Adulteration Using Spectroscopy-Based Sensors"

_foods, 2021, doi:10.3390/foods10040861_

Round 1

Reviewer 1 Report

This aim of the manuscript by Nychas et al. was to assess the potential of two spectroscopy-based sensors in detecting fraudulent minced meats substitution.

  1. The scope of this study finds good scientific and practical justification. It fits the scope of the journal. The manuscript is well designed and written. The references literature is appropriate.
  2. The study is performed systematically. The tools used are well selected and the conclusions drawn are complete in light of the obtained results. The chemometric qualitative analysis is well conducted, with multiple methods being applied and compared in performance and applicability.
  3. This work has been performed with care and the results are well exposed in the manuscript.

Considering the above, I suggest minor revision for this manuscript. I have some comments:

  1. For the Introduction, authors may find interesting these recent publications:

DOI: 10.3390/foods8020049

DOI: 10.3390/molecules24071402

  1. In the Table 3 – the appearance of NaN value (the outcome by division by 0) – should be explained
  2. It is confusing about Fluo data – there is no Fluo data in any Table, but in the Discussion one can notice that performance of Fluo was compared to MSI and Vis data.

Reviewer 2 Report

The manuscript “Detection of meat adulteration using spectroscopy-based sensors” presents an interesting study for the detection of adulteration in minced meatIn. For this purpose, they use different spectrochemical sensors and the SVM chemometric tool. In general terms, it seems to me to be a good approach for the detection of adulteration in meat without sample pre-treatment. This is the way forward for the detection of adulteration in all foods. They also use the figures of merit to compare the different models used. However, I have to make a few small points to clarify the content of the article.

My comments are as follows:

1.- Lines 59-62, the authors say “It is worth mentioning that the total number of requests concerning suspicion of fraud for the meat and poultry was greater than the number of requests for other food categories within the EU Administrative Assistance and Cooperation System 61 (AAC) in the annual report of 2019”. However, this report indicates that it is the third among the top 10 product categories in the AAC-FF in 2019.  Please correct this information.

2.- Line 89: (and vice versa). It makes sense to try to investigate the fraud of mixing chicken meat with pork. Chicken meat is cheaper than pork. Please indicate why you have done this study.

3.- Line 161-170: It would be desirable to include the parameter “Specificity”. This is also known as ‘true negative rate’ and shows the ratio of agreements of non-target class.
